# Prevention of 7-Ketocholesterol-Induced Overproduction of Reactive Oxygen Species, Mitochondrial Dysfunction and Cell Death with Major Nutrients (Polyphenols, ω3 and ω9 Unsaturated Fatty Acids) of the Mediterranean Diet on N2a Neuronal Cells

**DOI:** 10.3390/molecules25102296

**Published:** 2020-05-13

**Authors:** Aline Yammine, Thomas Nury, Anne Vejux, Norbert Latruffe, Dominique Vervandier-Fasseur, Mohammad Samadi, Hélène Greige-Gerges, Lizette Auezova, Gérard Lizard

**Affiliations:** 1Team Bio-peroxIL, Biochemistry of the Peroxisome, Inflammation and Lipid Metabolism’ EA 7270, University Bourgogne Franche-Comté, Inserm, 21000 Dijon, France; alineyammine5@gmail.com (A.Y.); thomas.nury@u-bourgogne.fr (T.N.); anne.vejux@u-bourgogne.fr (A.V.); norbert.latruffe@u-bourgogne.fr (N.L.); 2Bioactive Molecules Research Laboratory, Doctoral School of Sciences and Technologies, Faculty of Sciences, Lebanese University, Fanar, P.O. Box 90656 Jdeidet, Lebanon; hgreige@ul.edu.lb (H.G.-G.); lauezova@ul.edu.lb (L.A.); 3Team OCS, Institute of Molecular Chemistry of University of Burgundy (ICMUB UMR CNRS 6302), University Bourgogne Franche-Comté, 21000 Dijon, France; dominique.vervandier-fasseur@u-bourgogne.fr; 4LCPMC-A2, ICPM, Department of Chemistry, University Lorraine, Metz Technopôle, 57070 Metz, France; mohammad.samadi@univ-lorraine.fr

**Keywords:** apigenin, docosahexaenoic acid, eicosapentaenoic acid, 7-ketocholesterol, α-linolenic acid, Mediterranean diet, N2a cells, oleic acid, oxidative stress, quercetin, resveratrol.

## Abstract

The brain, which is a cholesterol-rich organ, can be subject to oxidative stress in a variety of pathophysiological conditions, age-related diseases and some rare pathologies. This can lead to the formation of 7-ketocholesterol (7KC), a toxic derivative of cholesterol mainly produced by auto-oxidation. So, preventing the neuronal toxicity of 7KC is an important issue to avoid brain damage. As there are numerous data in favor of the prevention of neurodegeneration by the Mediterranean diet, this study aimed to evaluate the potential of a series of polyphenols (resveratrol, RSV; quercetin, QCT; and apigenin, API) as well as ω3 and ω9 unsaturated fatty acids (α-linolenic acid, ALA; eicosapentaenoic acid, EPA; docosahexaenoic acid, DHA, and oleic acid, OA) widely present in this diet, to prevent 7KC (50 µM)-induced dysfunction of N2a neuronal cells. When polyphenols and fatty acids were used at non-toxic concentrations (polyphenols: ≤6.25 µM; fatty acids: ≤25 µM) as defined by the fluorescein diacetate assay, they greatly reduce 7KC-induced toxicity. The cytoprotective effects observed with polyphenols and fatty acids were comparable to those of α-tocopherol (400 µM) used as a reference. These polyphenols and fatty acids attenuate the overproduction of reactive oxygen species and the 7KC-induced drop in mitochondrial transmembrane potential (ΔΨm) measured by flow cytometry after dihydroethidium and DiOC_6_(3) staining, respectively. Moreover, the studied polyphenols and fatty acids reduced plasma membrane permeability considered as a criterion for cell death measured by flow cytometry after propidium iodide staining. Our data show that polyphenols (RSV, QCT and API) as well as ω3 and ω9 unsaturated fatty acids (ALA, EPA, DHA and OA) are potent cytoprotective agents against 7KC-induced neurotoxicity in N2a cells. Their cytoprotective effects could partly explain the benefits of the Mediterranean diet on human health, particularly in the prevention of neurodegenerative diseases.

## 1. Introduction

In Parkinson’s and Alzheimer’s disease, protein aggregation and mitochondrial dysfunction are two factors that promote oxidative stress, which is considered a major element in the evolution of these diseases [1]. In Parkinson’s disease, at the level of *substancia nigra pars compacta*, α-synuclein aggregates (Lewy bodies) induce a degeneration of dopaminergic neurons involving oxidative stress [1]. In Alzheimer’s disease, the increase in oxidative stress mediated by β-amyloid protein aggregates (senile plaques) in the hippocampus and cortex promotes excitotoxicity (pathological process by which nerve cells are damaged or killed by excessive stimulation by neurotransmitters) and synaptic degeneration leading to neurodegeneration [1]. Identifying natural or synthetic molecules that could prevent oxidative stress is therefore a part of therapeutic strategies for treatment of Parkinson’s and Alzheimer’s diseases. Among the molecules already identified are neuropeptides and natural antioxidants which have both antioxidant and/or anti-aggregation properties [1,2,3]. Furthermore, one of the consequences of oxidative stress is to promote the formation of lipid derivatives, including oxidized cholesterol derivatives (oxysterols) such as 7-ketocholesterol (7KC, also named 7-oxocholesterol) and 7β-hydroxycholesterol [4,5,6], which contribute to amplifying the oxidative stress that promotes inflammation and cell death, two other important components of neurodegeneration [7]. Preventing the toxicity of these oxysterols, particularly the predominantly formed 7KC, is also part of the pharmacological options for preventing neurodegeneration and more specifically Alzheimer’s disease [8]. Indeed, in post-mortem samples of the brain from Alzheimer’s disease patients, significant increases in 7KC have been identified in different areas from the frontal and occipital cortex [9]. In Parkinson’s disease, because of the oxidative stress present in the lesions, the contribution of 7KC is very probable [10]. Among the oxysterols associated with different diseases, 7KC is mainly increased in body fluids and lesions in age-related diseases (cardiovascular diseases, ocular diseases, Alzheimer’s disease, some cancers), in inflammatory bowel diseases and in some rare diseases (Niemann Pick’s disease, Smith Lemli Opitz syndrome and X-linked adrenoleukodystrophy (X-ALD)) [6,11]. Currently, various studies have shown a reduced risk of developing age-related diseases in people with a Mediterranean diet characterized by an important consumption in fruits and vegetables containing many polyphenols as well as fishes (sardines, mackerel, tuna) rich in ω3 and ω9 unsaturated fatty acids [12,13,14]. This suggests that nutrients present in the Mediterranean diet (polyphenols, fatty acids) could have cytoprotective effects against 7KC.

At the moment, only few data are available on the ability of polyphenols and fatty acids to prevent 7KC-induced cytotoxicity. The cytoprotective activities of polyphenols in the context of 7KC-induced toxicity involving oxidative stress that can lead to cell death, take into account their antioxidant activities linked to their chemical structures, but also their ability to act on the mitochondria by stimulating mitochondrial proliferation and activity [12,13]. Moreover, neurotrophic activities of several polyphenols are also described due to cytoprotective capacities (ability to prevent oxidative stress and cell death) associated with differentiating capacities on different types of cells [15]. Thus, on N2a cells, resveratrol (RSV) and apigenin (API) protect against hydrogen peroxide (H_2_O_2_)-induced cell death and also promote neuronal differentiation by stimulating the growth of neurites (dendrites and axons) [16,17]. Because of their neurotrophic properties, polyphenols such as RSV, QCT and API are therefore suitable molecules for counteracting neurodegeneration by preventing oxidative stress and stimulating neurogenesis [15,17]. When an oxysterols mixture containing 7α-hydroxycholesterol, 7β-hydroxycholesterol, 7-KC, cholesterol 5α,6α-epoxide, cholesterol 5β,6β-epoxide, cholestane-3β,5α,6β-triol and 25-hydroxycholesterol was used on human colon adenocarcinoma cells (Caco-2) [18] or human peripheral blood mononuclear cells [19], protective effects of cocoa bean shells containing a high level of epicatechin, and of olive oil polyphenols were observed, respectively. When differentiated murine PC12 cells and human neuroblastoma SH-SY5Y cells were treated with taxifolin (dihydroquercetin), it was also shown that 7KC-induced neuronal apoptosis was prevented [20]. On human retinal ARPE-19 cells, 7KC-induced cell death was also attenuated by RSV [21]. However, nothing is known on the ability of polyphenols to prevent 7KC-induced oxidative stress in murine N2a neuronal cells frequently used as pharmacological model in the context of neurodegenerative diseases [16,22]. So, we choose three polyphenols, which are widely represented in the Mediterrannean diet: RSV, QCT and API [12]. Among polyphenols, two classes are distinguished: Flavonoids and non-flavonoids [23]. Resveratrol (RSV; including trans-RSV which is biologically active) is a non-flavonoid; it is a member of the stilbenoid class of polyphenols; QCT (flavonol) and API (flavone) belong to the flavonoids class. RSV is mainly found in grapes, blackberries, peanuts and red wine. QCT and API are ubiquitously distributed in plant kingdom. QCT is present in lot of fruits and vegetables, and API in parsley, rosemary, celery and chamomile. Importantly, since polyphenols have the ability of interacting with the underlying pathomechanisms of several diseases associated with increased levels of 7KC, it was important to determine whether some of them (RSV, QCT and API) were able to prevent 7KC-induced cytotoxic effects, mainly reactive oxygen species (ROS) overproduction and mitochondrial dysfunction [8]. Although the ability of polyphenols to cross the blood-brain barrier is considered weak and remains still poorly understood [24,25], the intranasal administration of polyphenols (including RSV, QCT and API) has been proven effective to deliver these drugs to the brain while maintaining their activity [26].

As for fatty acids, the choice of these molecules to prevent the toxicity of 7KC takes into account their different and complementary properties from those of polyphenols. Compared to polyphenols, fatty acids do not by themselves have antioxidant properties. On the other hand, when they are added to cells on which oxidative stress is stimulated by some oxysterols, they can prevent oxidative stress as well as organelle dysfunction (mitochondria, lysosome and peroxisome) contributing to cell death [1]. Currently, only few lipids or lipid mixtures (oils) have been shown to be effective in preventing 7KC-induced ROS overproduction. These are α-tocopherol [27], ω3 and ω9 fatty acids (oleic acid (OA; C18:1 n-9) [28,29], and docosahexaenoic acid (DHA; C22:6 n-3)) [30,31]. In addition, several oils (argan oil, olive oil and milk thistle seed oil), which are rich in α-tocopherol and OA, have also shown cytoprotective effects against 7KC: they prevent organelle dysfunction, ROS overproduction and cell death [32]. While the cytoprotective mechanism of α-tocopherol can be explained at least in part by its ability to prevent the accumulation of 7KC in the lipid rafts [27,33], little is known on the cytoprotection of OA and DHA against 7KC. It can be assumed that they could both neutralize 7KC by esterification [34,35] and act by reducing oxidative stress and mitochondrial dysfunction leading to cell death [36,37]. Thus, in human U937 monocytic cells, while 7KC and 7β-OHC induce a type of death by apoptosis associated with oxidative stress and autophagic criteria, 7KC-oleate and 7β-OHC-oleate are not cytotoxic [35]. A major reason for the cytoprotective effects of fatty acids present in the Mediterranean diet could be the prevention of 7KC-induced plasma membrane destabilization which could lead to inactivation of the PDK1/Akt pathway resulting in the activation of GSK3 allowing the phosphorylation of Mcl-1 and the inhibition of sequestration in the cytoplasm of the pro-apoptotic molecules Bak and Bax; the latter could then interact with the mitochondrial membrane and contribute to inducing the drop in mitochondrial transmembrane potential (ΔΨm) activating apoptosis [38]. Furthermore, it is well recognized that DHA present in phospholipids in the sn-2 position can lead to the formation of the highly anti-apoptotic neuroprotectin D1 (NPD1) [39]. Furthermore, it is known that 7KC modifies the fluidity of the plasma membrane by intercalating between phospholipids [40]. Preventing 7KC-induced plasma membrane disorganization by using fatty acids associated with the Mediterranean diet is therefore a promising avenue to counter 7KC toxicity. The interest of fatty acids comes from their ability to pass the blood-brain barrier and accumulate in the brain; for DHA, several mechanisms associated with crossing the blood-brain barrier have been described [41]. As OA and DHA, which are present in significant amounts in the Mediterranean diet, can be used in functional foods and dietary supplements, it is therefore important to have more information on these molecules, as well as on the precursors of DHA (α-linolenic acid (ALA/C18:3 n-3) and eicosapentaenoic acid (EPA/C20:5 n-3)) which are present in significant amount in several Mediterranean fishes and oils.

In the present study, we evaluated and compared for the first time the cytoprotective effects of polyphenols (RSV, QCT and API) and fatty acids (OA, ALA, EPA, and DHA) on 7KC-treated N2a neuronal cells which are considered as a relevant model to characterize the toxicity of different compounds and to identify natural or synthetic molecules with cytoprotective activities that could give rise to pharmacological applications. The effects of these compounds were compared with those of α-tocopherol (the main component of Vitamin E constituted of four tocopherols and four tocotrienols) as the reference cytoprotective molecule. Our data show that nutrients (polyphenols, ω3 and ω9 unsaturated fatty acids) present at high amount in the Mediterranean diet have the ability to attenuate 7KC-induced ROS overproduction and cell death reinforcing the interest of the Mediterranean diet and some of its compounds for the prevention of certain age-related diseases such as neurodegenerative diseases.

## 2. Results

### 2.1. Effects of Polyphenols (Resveratrol, Quercetin, Apigenin), ω3 and ω9 Unsaturated Fatty Acids (α-Linolenic Acid, Eicosapentaenoic Acid, Docosahexaenoic Acid, Oleic Acid) and 7-Ketocholesterol on Cell Viability Evaluated with the Fluorescein Diacetate Assay

Whereas the polyphenols are known for their anti-oxidant properties, it is also well established that some of them have cytotoxic properties [42]. It is also known that several fatty acids can be toxic for several tissues, especially liver [43]. So, before to simultaneously treat N2a cells for 48 h with 7KC associated with polyphenols (RSV, QCT or API) or ω3 and ω9 unsaturated fatty acids (ALA, EPA, DHA and OA), it was important to evaluate the toxicity of these different compounds. Therefore, first of all, it was necessary to evaluate the cytotoxicity of studied compounds on N2a cells. To this end, the widely used fluorescein diacetate (FDA) assay, based on the measurement of esterase activity, was chosen to determine the cell viability. Comparatively to untreated cells, in a range of concentrations from 1.5 to 50 µM, significant cytotoxic effects were observed with trans-resveratrol (RSV) at 12.5, 25 and 50 µM (Figure 1A) and with QCT at 25 and 50 µM (Figure 1B) whereas no cytotoxic effects were found with API (Figure 1C). With RSV used at 6.25 µM, FDA activity was also decreased but the difference is not significant (Figure 1A); however, no decrease in FDA activity was observed with QCT and API used at this concentration (Figure 1B,C). In addition, comparatively to untreated cells, in a range of concentrations from 1.5 to 200 µM, significant cytotoxic effects were observed with ALA at 200 µM (Figure 2A), with EPA at 100, 150 and 200 µM (Figure 2B), with DHA at 100, 150 and 200 µM (Figure 2C), and with OA at 150 µM (Figure 2D). On the other hand, stimulating effects of the esterase activity revealed by the FDA assay were observed with ALA (25 µM) (Figure 2A), EPA (6.25 and 12.5 µM) (Figure 2B) and OA (25 µM) (Figure 2D). With 7KC used in a range of concentrations from 1.5 to 100 µM for 48 h, the 50% inhibiting concentration (IC50) was around 50 µM (Appendix A). Consequently, for further experiments, 7KC was therefore used at 50 µM, and among the concentrations of polyphenols and fatty acids chosen to assess cytoprotection, concentrations less than or equal to 6.25 µM, and 25 or 50 µM were used, respectively.

### 2.2. Evaluation with the Fluorescein Diacetate Assay of the Effects of Polyphenols (Resveratrol, Quercetin, Apigenin), ω3 and ω9 Unsaturated Fatty Acids (α-Linolenic Acid, Eicosapentaenoic Acid, Docosahexaenoic Acid, Oleic Acid) and α-Tocopherol on 7-Ketocholesterol-Induced Cytotoxicity

When 7KC (50 µM) was simultaneously incubated with the polyphenols (RSV, QCT or API; concentrations ≤6.25 µM, 48 h), the decrease of FDA positive cells observed under treatment with 7KC was strongly attenuated especially at 3.125 and 6.25 µM (Figure 1D–F). Similarly, when 7KC (50 µM) was simultaneously incubated with the ω9 and ω3 fatty acids (ALA, EPA, DHA and OA; concentrations ≤50 µM, 48 h), the decrease of FDA positive cells observed under treatment with 7KC was strongly attenuated especially at 12.5 and 25 µM whatever the fatty acid considered (Figure 2E–H). In the presence of α-tocopherol (400 µM), used as a reference to prevent 7KC-induced cell damages, cytoprotective effects were also found (Figure 1D–F; Figure 2E–H). Consequently, for further experiments, polyphenols have been used at 3.125 and 6.25 µM, and fatty acids at 12.5 and 25 µM.

### 2.3. Evaluation of the Effects of Polyphenols (Resveratrol, Quercetin, Apigenin), ω3 and ω9 Unsaturated Fatty Acids (α-Linolenic Acid, Eicosapentaenoic Acid, Docosahexaenoic Acid, Oleic Acid) and α-Tocopherol on 7-Ketocholesterol-Induced Reactive Oxygen Species (ROS) Overproduction

In N2a cells, 7KC (50 µM, 48 h) induces an overproduction of ROS which results in an increase in HE positive cells (Figure 3 and Figure 4). Since the most significant cytoprotective effects with the FDA assay were observed with polyphenols at 3.125 and 6.25 µM and with fatty acids at 12.5 and 25 µM, the ability of polyphenols and fatty acids to reduce 7KC-induced overproduction of ROS was investigated at these concentrations. By flow cytometry, after DHE staining, a strong increase in the percentage of HE positive cells (cells overproducing superoxide anions) was observed in the presence of 7KC (Figure 3 and Figure 4). The percentage of HE positive cells was significantly reduced by α-tocopherol (400 µM) used as positive control (Figure 3 and Figure 4). This percentage was also significantly reduced in the presence of polyphenols (RSV, QCT, API) (Figure 3) and fatty acids (ALA, EPA, DHA, OA) (Figure 4A,B) demonstrating that these compounds attenuate the oxidative stress induced by 7KC. When used singularly, polyphenols (QCT, API) and fatty acids (ALA, EPA, DHA, OA) as well as α-tocopherol have no effect on the production of ROS; on the other hand, comparatively to untreated (control) and vehicle-treated cells, RSV (6.25 µM) slightly but significantly increases this production (Figure 3 and Figure 4).

### 2.4. Evaluation of the Effects of ω3 and ω9 Unsaturated Fatty Acids (α-Linolenic Acid, Eicosapentaenoic Acid, Docosahexaenoic Acid, Oleic Acid) and α-Tocopherol on 7-Ketocholesterol-Induced Loss of Transmembrane Mitochondrial Potential and Cell Death

In N2a cells, 7KC (50 µM, 48 h) induces a loss of transmembrane mitochondrial potential (ΔΨm) as well as increased permeability of plasma membrane to PI which is considered to enter dead cells only (Figure 5 and Figure 6). Since the most significant cytoprotective effects with the FDA assay were observed with polyphenols at 3.125 and 6.25 µM and with fatty acids at 12.5 and 25 µM, the ability of polyphenols and fatty acids to prevent 7KC-induced loss of ΔΨm and cell death was investigated at these concentrations (Figure 5 and Figure 6). By flow cytometry, after DiOC_6_(3) staining, a strong increase in the percentage of DiOC_6_(3) negative cells (cells with low ΔΨm) was observed in the presence of 7KC (Figure 5 and Figure 6). In the presence of (QCT, API) and fatty acids (ALA, EPA, DHA and OA) as well as α-tocopherol no effect on ΔΨm was observed whereas with RSV (6.25 µM) a slight but significant increase in the percentage of DiOC_6_(3) negative cells (cells with low ΔΨm) was found (Figure 5 and Figure 6). The percentage of DiOC_6_(3) negative cells was significantly reduced by α-tocopherol (400 µM) used as positive control (Figure 5 and Figure 6). This percentage of DiOC_6_(3) negative cells was also significantly reduced in the presence of polyphenols (RSV, QCT, API) (Figure 5A–C) and fatty acids (ALA, EPA, DHA and OA) (Figure 6A–D) demonstrating that these compounds attenuate the loss of ΔΨm induced by 7KC.

In addition, by flow cytometry, after PI staining, a strong increase in the percentage of PI positive cells (cells with a loss of plasma membrane integrity considered as dead cells) was observed in the presence of 7KC (Figure 5 and Figure 6). The percentage of PI positive cells was also significantly reduced by α-tocopherol (400 µM) used as positive control (Figure 5 and Figure 6). This percentage of PI positive cells was also significantly reduced in the presence of polyphenols (RSV, QCT and API) (Figure 5D–F) and fatty acids (ALA, EPA, DHA and OA) (Figure 6E–H) demonstrating that these dietary compounds attenuate 7KC-induced cell death which is associated with plasma membrane alterations. Noteworthy, when used singularly, API (3.125 and 6.25 µM) and fatty acids (ALA, EPA, DHA and OA) (12.5 and 25 µM) as well as α-tocopherol (400 µM) have no effect on plasma membrane integrity whereas RSV and QCT (6.25 µM) slightly but significantly increase the % of PI positive cells indicating an effect of these molecules on plasma membrane integrity (Figure 5 and Figure 6).

## 3. Discussion 

Preventing the 7KC cytotoxicity by means of nutrients present in the Mediterranean diet (polyphenols, fatty acids) could be a means to counteract several diseases associated with oxidative stress leading to increased levels of 7KC mainly formed by cholesterol auto-oxidation [6,44]. In the present study, we showed that 7KC-induced cytotoxicity on murine neuronal N2a cells is characterized by ROS overproduction, alteration of plasma membrane, and drop of transmembrane mitochondrial potential (ΔΨm) leading to cell death. As the Mediterranean diet is rich in polyphenols, which are potent antioxidant molecules, we asked whether some of them such as RSV, QCT and API, could prevent the cytotoxic effects of 7KC. We also asked whether some fatty acids which are abundant in the Mediterranean diet (ALA, EPA, DHA and OA) and which are known to improve some brain functions, mainly cognition [45], were also able to prevent 7KC-induced cell death. Our results obtained with the polyphenols (RSV, QCT and API), and with ω3 and ω9 fatty acids (ALA, EPA, DHA and OA) on N2a cells show that these molecules are powerful cytoprotectors against 7KC-induced cytotoxicity: they strongly attenuate 7KC-induced ROS overproduction, alteration of plasma membrane, and loss of ΔΨm causing cell death.

As it was previously reported on microglial and glial cells (murine oligodendrocytes 158N and murine microglial BV-2 cells that 7KC triggers cell death [8,37], we asked whether similar effects were observed on N2a cells. In agreement with data obtained on 158N and BV-2 cells, our data demonstrate that 7KC (50 µM, 48 h of treatment) is a strong inducer of cell death on N2a cells. Whereas no significant cytotoxic effects were observed with 7KC used at 1.5625, 3.125, 6.25, 12.5 and 25 µM, significant cytotoxic effects were observed at 50 and 100 µM with the FDA assay (evaluating esterase activity)**.** In those conditions, after 48 h of culture of N2a cells with 7KC, the IC50 value of this oxysterol was 50 µM. So, the cell characteristics of 7KC-induced cell death were determined at this concentration. Also, in agreement with the abovementioned study on 158N and BV-2 cells, adding 7KC induced alterations of plasma membrane (decreased esterase activity measured by FDA, and increased permeability to PI) which could be at least in part the consequence of ROS overproduction measured with DHE. ROS overproduction in 7KC-treated human aortic smooth muscle cells and on human red blood cells is mediated by an upregulation of Nox-4, a ROS-generating NAD(P)H oxidase homologue [46], and is considered as a consequence of NADPH oxidase activation through a signaling mechanism including Rac GTPase and PKCζ [47], respectively. This important oxidative stress could also favor the drop of ΔΨm observed under treatment with 7KC. Our data confirm that ROS overproduction and loss of ΔΨm are general features associated with 7KC-induced cell death whatever the cells considered [6,8,37]. Thus, it is likely, that molecules which are able to prevent 7KC-induced toxicity in one cell type will also be effective in other cell types.

In the presence of 7KC, the highly conserved cell death characteristics from one type of cell to another may have important consequences for preventing the diseases associated with 7KC. It can thus be assumed that molecules capable of opposing the cytotoxicity of 7KC on N2a cells would also be capable to counteract the cytotoxicity of 7KC on other brain cells (glial and microglial cells) and on other cell types which can be exposed to high levels of 7KC (retinal and vascular cells). This hypothesis is partly validated by the use of α-tocopherol (400 µM) as a positive cytoprotective control in this study, which strongly reduces the cytotoxicity of 7KC not only on N2a but also on 158N oligodendrocytes and BV-2 microglial cells as well as many other cell types [8,37]. It can therefore be assumed that the cytoprotection observed on N2a with polyphenols (RSV, QCT, API at concentrations ≤6.25 µM) and fatty acids (ALA, EPA, DHA, OA at concentrations ≤50 µM) associated with the Mediterranean diet would present similar characteristics on other brain cells as those observed on N2a cells.

In order to inhibit or reduce 7KC-induced cell death and based on the signaling pathways of which have been described by Vejux et al. [37], we have chosen to use synthetic and natural molecules as well as mixtures of molecules [8]. To date, only few molecules have proved effective in opposing the cytotoxic effects of 7KC on nerve cells defined as oxiapoptophagy (OXIdative stress + APOPTOsis + autoPHAGY) based on data obtained on 158N murine oligodendrocytes and murine microglial BV-2 cells [48,49,50]. Among the synthetic molecules are two activators of the Nrf2 pathway, dimethyl fumarate (DMF) and its major metabolite, monomethyl fumarate (MMF), which have shown cytoprotective effects on 158N cells [51]. Some lipids, such as α-tocopherol, OA and DHA have also cytoprotective activities against 7KC: they prevent oxidative stress, mitochondrial dysfunction (loss of ΔΨm), peroxisomal changes, apoptosis and autophagy [8]. α-tocopherol, which prevents the accumulation of 7KC in the lipid rafts, inhibits the cascade of events leading to cell death [27,33].

Until now, there is no study on the effects of polyphenols (RSV, QCT and API) on 7KC-induced cell death on N2a neuronal cells. The present work demonstrates for the first time cytoprotective activities of some polyphenols (RSV, QCT and API) associated with the Mediterranean diet on 7KC-induced cell death and reinforces the interest of polyphenols to prevent 7KC-induced neurotoxicity. Indeed, on N2a cells, our data clearly show that RSV, QCT and API attenuate 7KC-induced ROS overproduction, plasma membrane alteration and drop of ΔΨm leading to cell death. The cytoprotective effects of RSV, QCT and API were studied at concentrations ≤6.25 µM because at higher concentrations a decrease of esterase activity was observed with RSV and QCT on N2a cells. These cytotoxic effects including decrease activity of esterase associated or not with ROS overproduction, loss of ΔΨm and/or increased plasma membrane permeability, which can affect cell viability, are probably related to the anti-tumor activities described with several polyphenols including RSV and QCT [52]. In contrast, API up to 50 µM does not induce cell death. Due to its lack of cytoxicity on N2a cells, API therefore appears to be a better candidate than RSV and QCT to prevent 7KC-induced cell death. Despite these differences in toxicity, at concentrations of 3.125 and 6.25 µM, all the three polyphenols, RSV, QCT and API, have similar cytoprotective effects against 7KC and are as efficient as α-tocopherol used at 400 µM.

Currently, as it is the case with polyphenols, the cytoprotective activities of ω3 and ω9 fatty acids (ALA, EPA, DHA and OA), present in large amount in the Mediterranean diet, on the neurotoxicity of 7KC are still not well known. We however previously reported on 158N murine oligodendrocytes and murine microglial BV-2 cells that 7KC-induced oxidative stress, mitochondrial dysfunction and cell death by oxiapoptophagy were strongly attenuated with DHA and OA as well as with argan, olive and milk thistle seed oils which are rich in α-tocopherol and OA [32]. In the present study, our data clearly show that ALA, EPA, DHA and OA also attenuate 7KC-induced ROS overproduction, plasma membrane alteration and drop of ΔΨm leading to cell death on N2a neuronal cells, thus bringing new elements reinforcing the interest of using ω3 and ω9 unsaturated fatty acids to prevent the neurotoxicity of 7KC. These cytoprotective effects of ω3 and ω9 fatty acids were found at concentrations (≤50 µM) higher than those identified with polyphenols (≤6.25 µM) but nevertheless lower than that obtained with α-tocopherol (400 µM). However, it is important to underline that at these concentrations ω3 and ω9 fatty acids do not show toxicity: They have no effects on esterase activity, ROS production and mitochondrial transmembrane potential (ΔΨm). Thus, in N2a cells, ω3 and ω9 unsaturated fatty acids (ALA, EPA, DHA, OA) but also polyphenols (RSV, QCT, API) are therefore more efficient natural molecules than α-tocopherol, considered until now as the best natural reference molecule to prevent 7KC toxicity. The ability of these polyphenols and fatty acids associated with the Mediterranean to prevent 7KC-induced cell damages reinforces the value of the Mediterranean diet and its associated nutrients in preventing neurodegeneration.

Since polyphenols (RSV, QCT, API) as well as ω3 and ω9 unsaturated fatty acids (ALA, EPA, DHA and OA) have similar cytoprotective effects on 7KC-induced toxicity and reduce plasma membrane alteration, ROS overproduction and drop in ΔΨm, pharmacological targets that are otherwise common but lead to the same effects should be sought. Based on the cytoprotective effects observed with natural and synthetic molecules [8], RSV, QCT and API as well as ALA, EPA, DHA, and OA could reduce the overproduction of ROS via the Nrf2 pathway and by acting on the activation of NADPH-oxidase. Since (RSV, QCT, API), and ω3 and ω9 fatty acids can activate the nuclear Liver X Receptors (LXRs) [53] and Peroxisome Proliferator-Activated Receptors (PPARs) [54], respectively, cytoprotection resulting from gene activation, which could be evaluated by transcriptomic analysis, is also possible. In order to link the cytoprotective activity of polyphenols and fatty acids from the plasma membrane to the mitochondria, an action at the PI3-K/PDK1/Akt/GSK3 pathway is possible. Preserving the functionality of this pathway makes it possible to link plasma membrane signaling to mitochondrial activity. This hypothesis has moreover been validated on U937 cells treated with 7KC associated or not with α-tocopherol [27,38].

In conclusion, on N2a cells, 7KC induces cell damages (oxidative stress and mitochondrial dysfunction) associated with neurodegeneration on N2a cells which are strongly attenuated by polyphenols (RSV, QCT and API) and fatty acids (ALA, EPA, DHA and OA) present in large amounts in the Mediterranean diet. These data provide new evidence in favor of 7KC as a potential risk factor for neurodegeneration and shows the capacity of some nutrients (polyphenols, fatty acids) as well as α-tocopherol to prevent 7KC-induced cytotoxicity: ROS overproduction, loss of ΔΨm and cell death associated with enhanced plasma membrane permeability. This study also underlines the chemopreventive effects of phytochemicals against cell dysfunction caused by oxysterols [55]. On the other hand, in vitro, in order to enhance the cytoprotection, a combined use of these nutrients can be envisaged. In vivo, due to the likely and rapid degradation of these nutrients (elimination in the gastrointestinal tract, short plasma half-life) before reaching their target, various micro- and nano-encapsulation strategies protecting these nutrients without altering their cytoprotective activities could be considered for further studies [56,57]. Thus, when QCT is nanoencapsulated, it has been described to decrease the inflammation induced by various oxysterols on SH-SY5Y human neuronal cells [58]. As mitochondrial and ROS overproduction is a general feature of 7KC-induced cell death whatever the cells considered, it may be advisable to micro- or nano-encapsulate these nutrients in order to promote access to their therapeutic targets. In addition, the ability of polyphenols and fatty acids to strongly reduce the 7KC-induced drop in ΔΨm indicates that these nutrients are potentially suitable molecules for mitotherapy [59]. Associated a targeted mitotherapy using functionalized nanoparticles (Targeted Organel Nano-therapy: TORN therapy) [60,61] containing polyphenols as well as ω3 and/or ω9 unsaturated fatty acids could thus be envisaged to treat diseases associated with increased 7KC-levels.

## 4. Materials and Methods

### 4.1. Cell Culture and Treatments

The murine neuro-2a (N2a) neuroblastoma cell line (Ref: CCL-131; ATCC, Manassas, VA, USA) is maintained in Dulbecco’s modified Eagle medium (DMEM, Lonza, Amboise, France) containing 10% (*v/v*) of heat-inactivated fetal bovine serum (FBS) (Pan Biotech, Aidenbach, Germany) (30 min, 56 °C) and 1% (*v/v*) of penicillin (100 U/mL) / streptomycin (100 mg/mL) (Pan Biotech). They were incubated at 37 °C in a humidified atmosphere (5% CO_2_, 95% air) and passaged twice a week. The cells were seeded, for the different experimental conditions, at a density of 1.2 × 10^5^ cells per well containing 1 mL of culture medium with 10% FBS in 6-well plates (FALCON, Becton Dickinson, Le Pont de Claix, France) or in Petri dishes at 30.000 cells/cm^2^ (100 mm diameter) in order to assess the ability of polyphenols (trans-resveratrol (RSV), quercetin (QCT) and apigenin (API)) and fatty acids (α-linolenic acid (ALA/C18:3 n-3); eicosapentaenoic acid (EPA/C20:5 n-3); docosahexaenoic acid (DHA, C22:6 n-3); oleic acid (OA, C18:1 n-9)) to counteract the cytotoxicity induced by 7-ketocholesterol (7KC). 7KC (Ref: C2394), the polyphenols (RSV, QCT and API) and the fatty acids (ALA, DHA and OA) were from Sigma-Aldrich (St Quentin-Fallavier, France) whereas EPA was from Enzo Life Sciences (Villeurbanne, France). The stock solution of 7KC (800 µg/mL; 2 mM) was prepared in a mixture of absolute ethanol and culture medium (0.04, 0.96, *v/v*) as previously described [27]. The stock solutions of polyphenols were prepared as follows: RSV at 50 mM in absolute ethanol (EtOH; Carlo Erba Reagents, Val de Reuil, France), whereas dimethyl sulfoxide (DMSO; Sigma-Aldrich) was used as vehicle to dissolve QCT and API prepared at 50 mM. Fatty acids stock solutions were prepared at 50 mM (ALA, DHA, OA) or 200 mM (EPA) in absolute ethanol and stored at −20 °C. After 24 h of culture, the culture medium was removed and the cells were further incubated for an additional 48 h period of time with or without polyphenols or fatty acids associated or not with 7KC. Polyphenols (RSV, QCT and API) or fatty acids (ALA, EPA, DHA and OA) were used at concentrations ranging from 1.5 to 50 µM and from 1.5 to 200 µM, respectively, to determine their effects on cell viability. When associated with 7KC, polyphenols (RSV, QCT and API) and fatty acids (ALA, EPA, DHA and OA) were used at concentrations which have no impact on cell viability ranging from 1.5 to 6.25 µM and from 6.25 to 50 µM, respectively. The choice of the concentration of 7KC is based on the viability tests performed in this study which show that 50 µM is the 50% inhibiting concentration (IC50) for N2a cells (Appendix A). α-tocopherol (Sigma-Aldrich) was used as a positive control for cytoprotection; the α-tocopherol solution was prepared at 80 mM in absolute ethanol and diluted in the culture medium to obtain a 400 μM final concentration. α-tocopherol was used at the highest non-cytotoxic concentration (400 µM) able to prevent 7KC-induced apoptosis [27].

### 4.2. Quantification of Cell Viability by the Fluorescein Diacetate (FDA) Assay 

N2a cells viability was assessed using a lipophilic fluorochrome, fluorescein diacetate (FDA) (Sigma-Aldrich). In the presence of intracellular esterases of living cells, the non-fluorescent FDA is transformed into a green fluorescent metabolite “fluorescein” [16,22]. At the end of the treatment, cells were incubated in the dark with FDA (15 µg/mL, 5 min, 37 °C), washed twice with phosphate buffer saline (PBS 0.1 M, pH 7.4), and then lysed with 10 mM Tris-HCl solution containing 1% sodium dodecyl sulfate (SDS). Fluorescence intensity of fluorescein was measured with excitation at 485 nm and emission at 528 nm using a microplate reader (Sunrise spectrophotometer, TECAN, Lyon, France) in order to quantify living cells. All assays were performed in at least four independent experiments and realized in triplicate. The results were expressed as % of control: (Fluorescence (assay) × 100)/Fluorescence (control).

### 4.3. Flow Cytometric Evaluation of Reactive Oxygen Species (ROS) Production Using Dihydroethidium Staining

ROS overproduction, including the intracellular superoxide anion (O_2_^●−^), was detected by dihydroethidium staining (DHE). DHE is a non-fluorescent probe capable of diffusing through the cell membranes; it is rapidly oxidized by ROS to fluorescent hydroethidium (HE) that exhibits an orange/red fluorescence (λ Ex max = 488 nm; λ Em max = 575nm). The HE intercalates at the DNA base level [62]. The stock solution of DHE (Invitrogen/Thermo Fisher Scientific, Courtaboeuf, France) was prepared in DMSO at a concentration of 10 mM and was subsequently used on the cells at a final concentration of 2 μM. N2a cells incubated for 48 h with or without 7KC in the presence or absence of polyphenols (RSV, QCT or API) or fatty acids (ALA, EPA, DHA, or OA) were trypsinized, washed and then suspended in 1 mL of 1X PBS containing DHE (2 μM). After 15 min of incubation at 37 °C, the analysis of stained cells was carried out by flow cytometry on a BD Accuri™ C6 flow cytometer. The fluorescence of HE was collected through a 580 nm band pass filter. Ten thousand cells were acquired per sample and the fluorescence was quantified on a logarithmic scale. The data were analyzed with FlowJo software v10.6.2 (Tree Star Inc., Ashland, OR, USA). The percentage of ROS producing cells corresponds to the percentage of HE positive cells. At least three independent experiments were realized.

### 4.4. Flow Cytometric Measurement of Transmembrane Mitochondrial Potential Using DiOC_6_(3)

The measurement of transmembrane mitochondrial potential (ΔΨm) is widely used to characterize mitochondrial metabolic state and cell death. The decrease in ΔΨm is a sign of a depolarized mitochondria with a poor ability to accumulate lipophilic cationic probes such as 3, 3′-dihexyloxacarbocyanine iodide DiOC_6_(3) (λ Ex max = 484 nm; λ Em max = 501 nm) used in flow cytometry. DiOC_6_(3) accumulates in the mitochondrial matrix as a function of the potential difference: this probe accumulates all the more in the mitochondria with a normal ΔΨm (living cells) compared to the mitochondria with a low ΔΨm (dying and dead cells) which will cause a decrease in green fluorescence collected through a 520 ± 10 nm band pass filter. To evaluate the effect on ΔΨm of the different compounds tested at different concentrations, the adherent and non-adherent cells were pooled after 48 h of treatment, washed and stained with a solution of DiOC_6_(3) (Invitrogen/Thermo Fisher Scientific) at 40 nM for 15 min at 37 °C. At the end of the incubation time, the fluorescence associated with DiOC_6_(3) was immediately measured on a BD Accuri™ C6 flow cytometer. Ten thousand cells were acquired per sample and the fluorescence was quantified on a logarithmic scale. Data were analyzed with FlowJo software (Tree Star Inc.). The percentage of cells with depolarized mitochondria corresponds to the percentage of DiOC_6_(3) negative cells. At least three independent experiments were realized.

### 4.5. Flow Cytometric Evaluation of Plasma Membrane Permeability and Cell Death Using Propidium Iodide

Propidium iodide (PI) is a hydrophilic fluorescent agent intercalating nucleic acids (DNA; RNA) which produces a red/orange fluorescence when it is excited by a blue light (488 nm). It is used to quantify cell viability and plasma membrane permeability: it only stains the nucleic acids of dead cells that have lost their cytoplasmic membrane integrity [63,64]. The stock solution of PI was prepared in milliQ water at a concentration of 1 mg/mL. N2a cells (adherent and nonadherent cells) were stained with a PI solution at 1 μg/mL in 1X PBS for 5 min at 37 °C in order to assess their mortality after a treatment of 48 h with polyphenols (RSV, QCT or API) or with fatty acids (ALA, EPA, DHA, and OA) associated or not with 7KC. The cells were analyzed on a BD Accuri™ C6 flow cytometer and the fluorescence of PI was selected on a 630 nm long pass filter. Ten thousand cells were acquired per sample and the fluorescence was quantified on a logarithmic scale. Data were analyzed with FlowJo software (Tree Star Inc.). The percentage of dead cells corresponds to the percentage of PI positive cells. At least three independent experiments were realized.

### 4.6. Statistical Analysis

Statistical analyses were done using XLSTAT software 2020.1 (Microsoft, St. Redmond, WA, USA). Data were expressed as mean ± standard deviation (SD); data were considered statistically different (Mann-Whitney test) at a *P*-value of 0.05 or less.

## Figures and Tables

**Figure 1 molecules-25-02296-f001:**
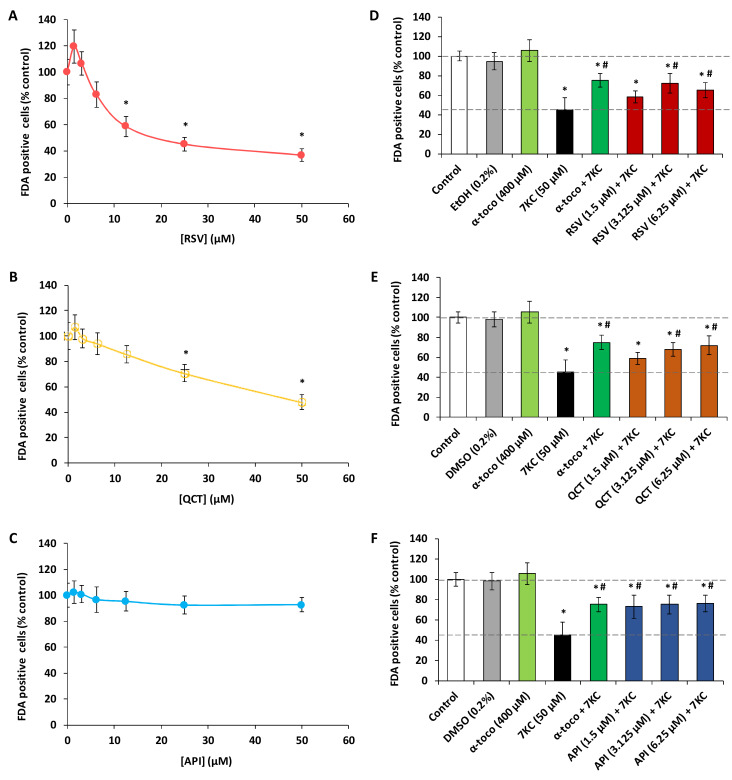
Evaluation with the fluorescein diacetate (FDA) assay of the effects of polyphenols (resveratrol, quercetin and apigenin) with and without 7-ketocholesterol on cell viability of N2a cells. Murine neuroblastoma N2a cells, previously cultured for 24 h, were further cultured for 48 h with or without resveratrol (RSV), quercetin (QCT), apigenin (API) in the presence or absence of 7-ketocholesterol (7KC; 50 µM). Polyphenols concentrations used alone range from 1.5 to 50 µM (**A**–**C**). Polyphenols were used at concentrations ≤6.25 µM when associated with 7KC (50 µM) (**D**–**F**). α-tocopherol (400 µM) was used as positive reference to prevent 7KC-induced cell death. The results are expressed in percentages relatively to the control (untreated cells). Data obtained with the FDA assay are shown. Data shown are expressed as mean ± standard deviation (SD) of four independent experiments performed in triplicate. Significance of the differences between control (untreated cells) and RSV-, QCT-, API-, α-toco or 7KC-treated cells; Mann Whitney test: * *P* < 0.05 or less. Significance of the differences between 7KC-treated cells and (7KC + (RSV, QCT, API or α-toco))-treated cells; Mann Whitney test: # *P* < 0.05 or less. No significant differences were found between control and vehicle-treated cells (ethanol (EtOH) and DMSO).

**Figure 2 molecules-25-02296-f002:**
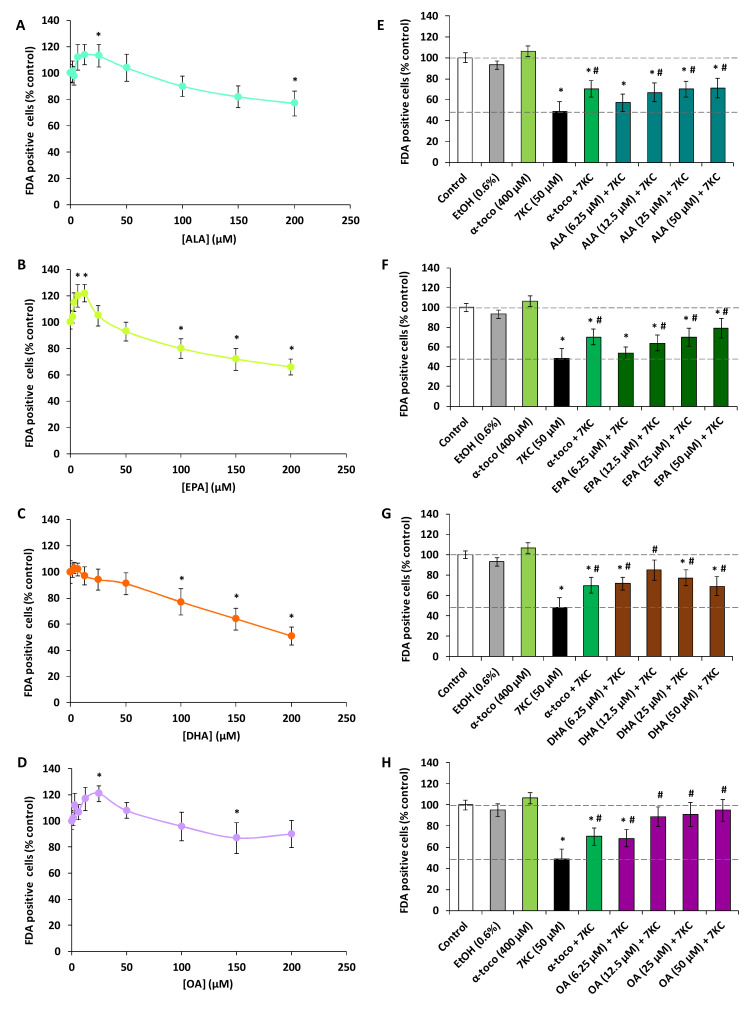
Evaluation with the fluorescein diacetate (FDA) assay of the effects of ω3 and ω9 unsaturated fatty acids (α-linolenic acid, eicosapentaenoic acid, docosahexaenoic acid and oleic acid) with and without 7-ketocholesterol on cell viability of N2a cells. Murine neuroblastoma N2a cells, previously cultured for 24 h, were further cultured for 48 h with or without α-linolenic acid (ALA), eicosapentaenoic acid (EPA), docosahexaenoic acid (DHA) or oleic acid (OA) in the presence or absence of 7-ketocholesterol (7KC; 50 µM). Fatty acids concentrations used alone range from 1.5 to 200 µM (**A**–**D**). Fatty acids were used at concentrations ≤50 µM when associated with 7KC (50 µM) (**E**–**H**). α-tocopherol (α-toco; 400 µM) was used as positive reference to prevent 7KC-induced cell death. The results are expressed in percentages relatively to the control (untreated cells). Data obtained with the FDA assay are shown. Data shown are expressed as mean ± standard deviation (SD) of four independent experiments performed in triplicate. Significance of the differences between control (untreated cells) and ALA-, EPA-, DHA-, OA-, α-toco or 7KC-treated cells; Mann Whitney test: * *P* < 0.05 or less. Significance of the differences between 7KC-treated cells and (7KC + (ALA, EPA, DHA, OA or α-toco))-treated cells; Mann Whitney test: # *P* < 0.05 or less. No significant differences were found between control and vehicle-treated cells (Ethanol (EtOH)).

**Figure 3 molecules-25-02296-f003:**
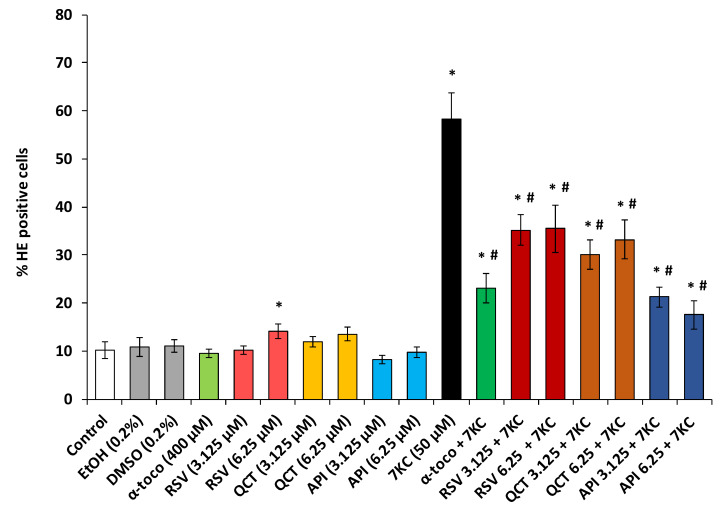
Effect of polyphenols (resveratrol, quercetin, apigenin) with and without 7-ketocholesterol on reactive oxygen species (ROS) overproduction. N2a cells, previously cultured for 24 h, were further cultured for 48 h with or without 7-ketocholesterol (7KC, 50 μM) in the presence or absence of polyphenols: resveratrol (RSV), quercetin (QCT) or apigenin (API) used at a concentration of 3.125 and/or 6.25 µM. α-tocopherol (α-toco; 400 µM) was used as positive reference to prevent 7KC-induced cell death. ROS overproduction was measured by flow cytometry after staining with dihydroethidine (DHE) and evaluated by the percentage of HE positive cells. Data shown are mean ± standard deviation (SD) of three independent experiments conducted in triplicate. Significance of the differences between control (untreated cells) and RSV-, QCT-, API-, α-toco- or 7KC-treated cells; Mann Whitney test: * *P* < 0.05 or less. Significance of the differences between 7KC-treated cells and (7KC + (RSV, QCT, API or α-toco))-treated cells; Mann Whitney test: # *P* < 0.05 or less. No significant differences were found between control and vehicle-treated cells (Ethanol (EtOH) and DMSO).

**Figure 4 molecules-25-02296-f004:**
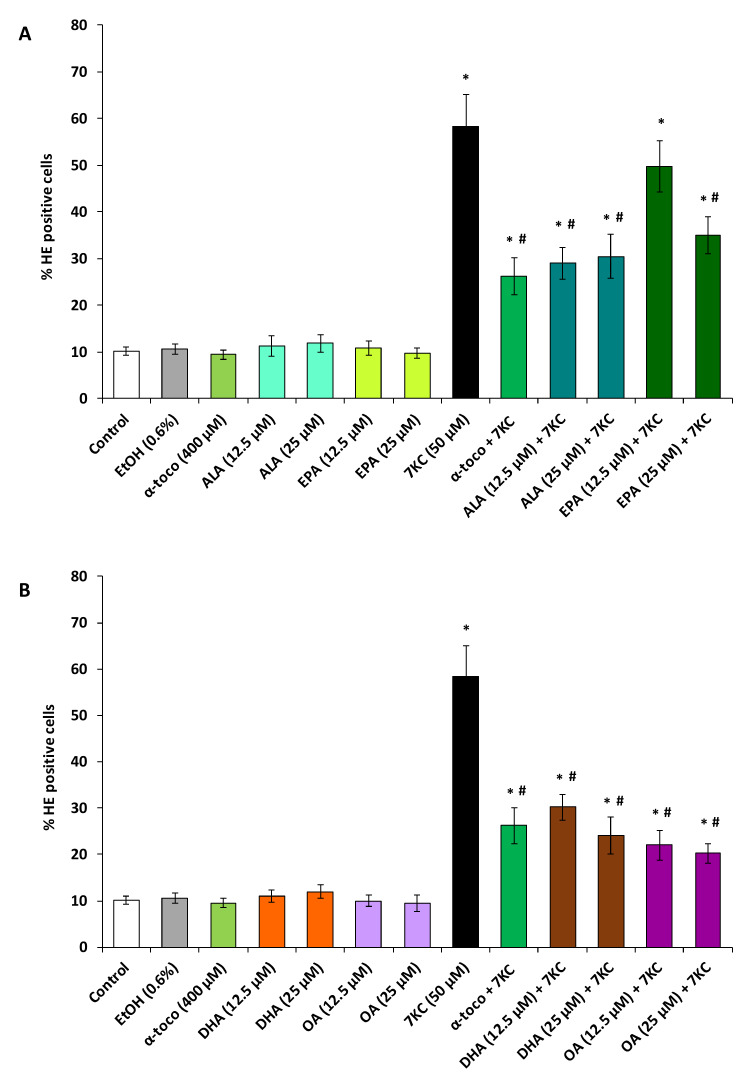
Effect of ω3 and ω9 unsaturated fatty acids (α-linolenic acid, eicosapentaenoic acid, docosahexaenoic acid and oleic acid) with and without 7-ketocholesterol on reactive oxygen species (ROS) overproduction. N2a cells, previously cultured for 24 h, were further cultured for 48 h with or without 7-ketocholesterol (7KC, 50 μM) in the presence or absence of α-linolenic acid (ALA), eicosapentaenoic acid (EPA), docosahexaenoic acid (DHA) or oleic acid (OA) used at a concentration of 12.5 and 25 µM. α-tocopherol (α-toco; 400 µM) was used as positive reference to prevent 7KC-induced cell death. ROS overproduction was measured by flow cytometry after staining with dihydroethidine (DHE) and evaluated by the percentage of HE positive cells (**A**): data obtained with ALA and EPA are shown; (**B**): data obtained with DHA and OA are shown). Data shown are mean ± standard deviation (SD) of three independent experiments conducted in triplicate. Significance of the differences between control (untreated cells) and ALA-, EPA-, DHA-, OA-, α-toco- or 7KC-treated cells; Mann Whitney test: * *P* < 0.05 or less. Significance of the differences between 7KC-treated cells and (7KC + (ALA, EPA, DHA, OA or α-toco))-treated cells; Mann Whitney test: # *P* < 0.05 or less. No significant differences were found between control and vehicle-treated cells (Ethanol (EtOH)).

**Figure 5 molecules-25-02296-f005:**
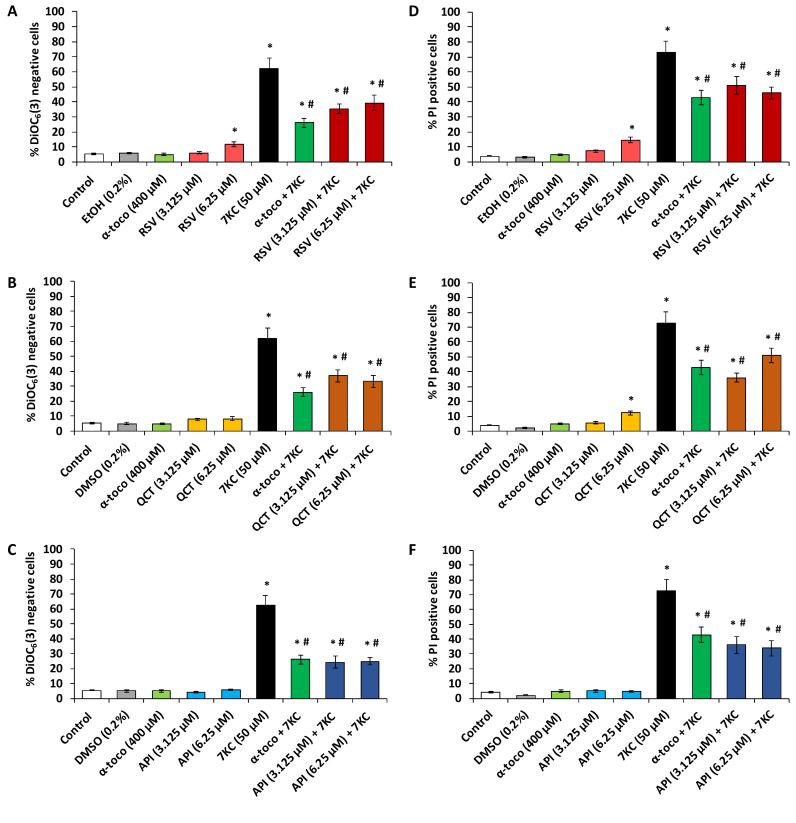
Effect of polyphenols (resveratrol, quercetin, apigenin) with and without 7-ketocholesterol on mitochondrial transmembrane potential (ΔΨm) and plasma membrane permeability. N2a cells, previously cultured for 24 h, were further cultured for 48 h with or without 7-ketocholesterol (7KC, 50 μM) in the presence or absence of polyphenols: resveratrol (RSV), quercetin (QCT) or apigenin (API) used at a concentration of 3.125 and/or 6.25 µM. α-tocopherol (α-toco; 400 µM) was used as positive reference to prevent 7KC-induced cell death. Loss of transmembrane mitochondrial potential (ΔΨm) was measured by flow cytometry after staining with DiOC_6_(3) and evaluated by the percentage of DiOC_6_(3) negative cells (**A**–**C**). Plasma membrane permeability was measured by flow cytometry after staining with propidium iodide (PI): for each assay, the percentage of PI positive cells was determined (**D**–**F**). Data shown are mean ± standard deviation (SD) of three independent experiments conducted in triplicate. Significance of the differences between control (untreated cells) and RSV-, QCT-, API-, α-toco- or 7KC-treated cells; Mann Whitney test: * *P* < 0.05 or less. Significance of the differences between 7KC-treated cells and (7KC + (RSV, QCT, API or α-toco))-treated cells; Mann Whitney test: # *P* < 0.05 or less. No significant differences were found between control and vehicle-treated cells (Ethanol (EtOH) and DMSO).

**Figure 6 molecules-25-02296-f006:**
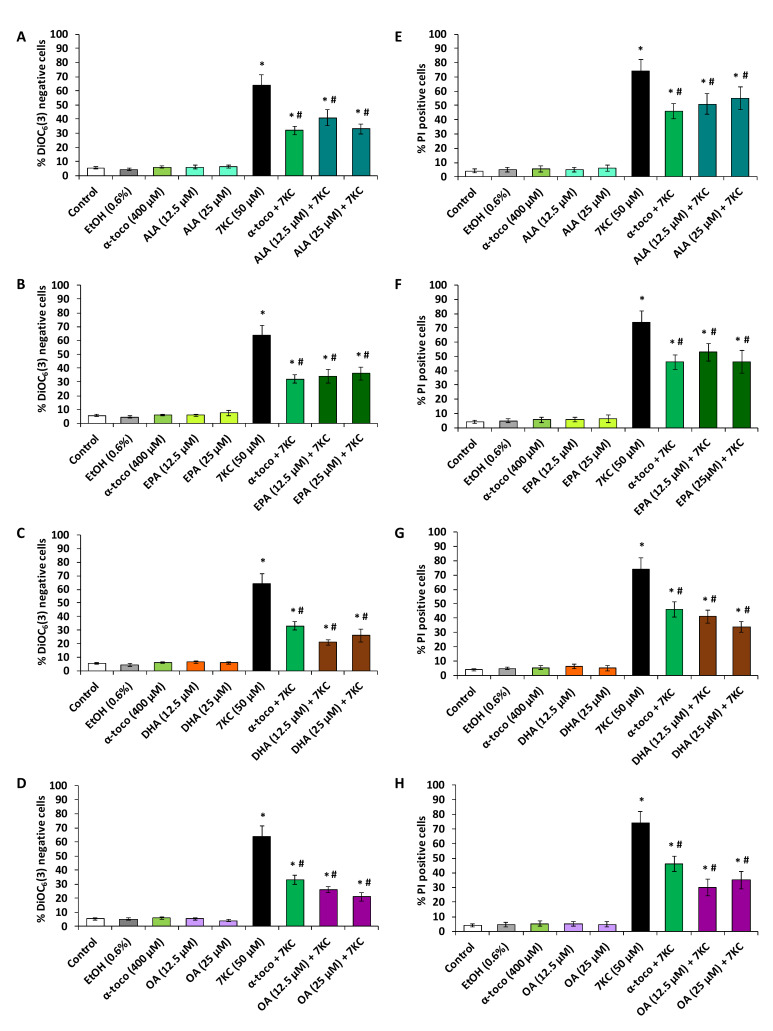
Effect of ω3 and ω9 unsaturated fatty acids (α-linolenic acid, eicosapentaenoic acid, docosahexaenoic acid and oleic acid) with and without 7-ketocholesterol on mitochondrial transmembrane potential (ΔΨm) and plasma membrane permeability. N2a cells, previously cultured for 24 h, were further cultured for 48 h with or without 7-ketocholesterol (7KC, 50 μM) in the presence or absence of α-linolenic acid (ALA), eicosapentaenoic acid (EPA), docosahexaenoic acid (DHA) or oleic acid (OA) used at a concentration of 12.5 and 25 µM. α-tocopherol (α-toco; 400 µM) was used as positive reference to prevent 7KC-induced cell death. Loss of transmembrane mitochondrial potential (ΔΨm) was measured by flow cytometry after staining with DiOC_6_(3) and evaluated by the percentage of DiOC_6_(3) negative cells (**A**–**D**). Plasma membrane permeability was measured by flow cytometry after staining with propidium iodide (PI): for each assay, the percentage of PI positive cells was determined (**E**–**H**). Data shown are mean ± standard deviation (SD) of three independent experiments conducted in triplicate. Significance of the differences between control (untreated cells) and ALA-, EPA-, DHA-, OA-, α-toco- or 7KC-treated cells; Mann Whitney test: * *P* < 0.05 or less. Significance of the differences between 7KC-treated cells and (7KC + (ALA, EPA, DHA, OA or α-toco))-treated cells; Mann Whitney test: # *P* < 0.05 or less. No significant differences were found between control and vehicle-treated cells (Ethanol (EtOH)).

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
