# Peer review of "Prevention of 7-Ketocholesterol-Induced Overproduction of Reactive Oxygen Species, Mitochondrial Dysfunction and Cell Death with Major Nutrients (Polyphenols, ω3 and ω9 Unsaturated Fatty Acids) of the Mediterranean Diet on N2a Neuronal Cells"

_molecules, 2020, doi:10.3390/molecules25102296_

Round 1
Reviewer 1 Report
The MS describes the effect of some polyphenols and FA on induced toxicity by ROS production of 7KC on N2a neuronal cells. The MS is well written. The experimental designed was clearly stated and performed. There are some minor issues described below that should be attended. I just suggest add to the introduction and discussion sections as well, a description of the differences of each tested compound used as an antioxidants (i.e. polyphenols and FA), all this since important differences were found in all analyses performed. For example, according to your own results API showed better results than RSV and QCT on HE positive cells, mitochondrial transmembrane potential, and plasma membrane permeability. Differences were also observed between all FA tested, e.g. OA was more effective than ALA, EPA and DHA. You tested different kind of PO and FA with important chemical differences that should be addressed in the introduction section, but also and more important, properly discussed. This will greatly improve the MS.
All comments and suggestions should be attended before considering this MS for publication in this Journal.
Specific comments:
Line 48: Reduce the number of keywords
Line 58 to 60: font size
Line 95: add the word “resveratrol” after their acronym
Line 221: font size
Line 261: why viability cells are over a 100% and effect was significant in lower concentrations? This was indicated by asterisks.
Line 412: font size
Line 422 to 423: These asseverations needs to be tested and proved
Line 468: font size
Author Response
ANSWERS REVIEWER 1
Prevention of 7-ketocholesterol-induced overproduction of reactive oxygen species, mitochondrial dysfunction and cell death with major nutrients (polyphenols, ω3 and ω9 unsaturated fatty acids) of the Mediterranean diet on N2a neuronal cells –YAMMINE A et al.
Reviewer 1 (the answers are underlined in yellow)
The MS describes the effect of some polyphenols and FA on induced toxicity by ROS production of 7KC on N2a neuronal cells. The MS is well written. The experimental designed was clearly stated and performed. There are some minor issues described below that should be attended.
Question: I just suggest add to the introduction and discussion sections as well, a description of the differences of each tested compound used as an antioxidants (i.e. polyphenols and FA), all this since important differences were found in all analyses performed. For example, according to your own results API showed better results than RSV and QCT on HE positive cells, mitochondrial transmembrane potential, and plasma membrane permeability. Differences were also observed between all FA tested, e.g. OA was more effective than ALA, EPA and DHA. You tested different kind of PO and FA with important chemical differences that should be addressed in the introduction section, but also and more important, properly discussed. This will greatly improve the MS.
Answer: this has been done; important modifications have been realized in the introduction as well as in the discussion. Additional information is provided on the properties of polyphenols and fatty acids and on their potential mechanisms to prevent 7KC-induced cell death. Several details are provided allowing the comparaison of polyphenols and fatty acids. New references have been added.
- Specific comments:
Question: Line 48; Reduce the number of keywords
Answer: this has been done
Question: Line 58 to 60; font size
Answer: the modification has been done
Question: Line 95; add the word “resveratrol” after their acronym
Answer: this has been done
Question: Line 221; font size
Answer: the modification has been done
Question: Line 261; why viability cells are over a 100% and effect was significant in lower concentrations? This was indicated by asterisks.
Answer: In order to avoid confusion instead of writing % of viable cells, it is now written % of FDA positive cells (Figures 1 and 2 have been consequently modified)
Question: Line 412; font size
Answer: the modification has been done
Question: Line 422 to 423; These asseverations needs to be tested and proved
Answer: we agree, the sentence have been modified and the paragraph rewritten.
Question: Line 468; font size
Answer: the modification has been done
Reviewer 2 Report
The results presented in the manuscript are consistent with its title; however the authors overinterpreted the findings in the discussion.
First of all, it is unjustified to suggest the use of tested polyphenols against AD and X-ALD since RSV, QCT, and API have not been shown to cross the BBB. In line 101, it was only written that "Importantly, since polyphenols have the ability to cross the blood brain barrier [22, 23]”. Of note, reference 22 supports only transport of QCT across the BBB, while reference 23 dealing with in silico predictions and in vitro transport studies across BBB endothelial cells do not even refer to the other two polyphenols. Moreover, the demonstration of cytoprotective properties in cell line models in no way confirms the efficacy of tested compounds against the indicated neurodegenerative diseases. In this context, referring to nonencapsulated formulation is also unclear.
Lines 102-105 and 485-490, Indicating the chemopreventive capacity of tested compounds based on their cytoprotective effect in neuroblastoma (N2a) cells, is also unclear.
In addition, the authors should comment on the potentially adverse effect of RSV alone at a concentration of 6.25 mM on ROS overproduction, ΔΨm, and plasma membrane permeability.
Author Response
ANSWERS REVIEWER 2
Prevention of 7-ketocholesterol-induced overproduction of reactive oxygen species, mitochondrial dysfunction and cell death with major nutrients (polyphenols, ω3 and ω9 unsaturated fatty acids) of the Mediterranean diet on N2a neuronal cells –YAMMINE A et al.
Reviewer 2 (The answers are underlined in green)
The results presented in the manuscript are consistent with its title; however the authors overinterpreted the findings in the discussion.
Question: First of all, it is unjustified to suggest the use of tested polyphenols against AD and X-ALD since RSV, QCT, and API have not been shown to cross the BBB. In line 101, it was only written that "Importantly, since polyphenols have the ability to cross the blood brain barrier [22, 23]”. Of note, reference 22 supports only transport of QCT across the BBB, while reference 23 dealing with in silico predictions and in vitro transport studies across BBB endothelial cells do not even refer to the other two polyphenols. Moreover, the demonstration of cytoprotective properties in cell line models in no way confirms the efficacy of tested compounds against the indicated neurodegenerative diseases. In this context, referring to nonencapsulated formulation is also unclear.
Answer: The question of BBB is a very important point and we agree with the different remarks of the reviewer. The different parts of the manuscript concerning the passage of the BBB either for the polyphenols or for the fatty acids have been rewritten and new recent references have been added. These points are discussed and nuanced. We also agree with the last sentence of the remark of the reviewer and the manuscript has been improved on this point; the interest to use nanoparticles, especially for the polyphenols, has been clarified in the conclusion.
Lines 102-105 and 485-490, Indicating the chemopreventive capacity of tested compounds based on their cytoprotective effect in neuroblastoma (N2a) cells, is also unclear.
Answer: the sentences have been re-written.
In addition, the authors should comment on the potentially adverse effect of RSV alone at a concentration of 6.25 mM on ROS overproduction, ΔΨm, and plasma membrane permeability.
Answer: the adverse effects of RSV is now clearly indicated in the results and it is also discussed.
Round 2
Reviewer 2 Report
The authors addressed most comments, and the manuscript has improved as a result. However, there is still a confusing description of results.
The commentary that "Noteworthy, when used singularly, polyphenols (QCT, API, RSV) and fatty acids (ALA, 397 EPA, DHA, OA) as well as α-tocopherol have no effect on plasma membrane integrity (Figures 5-6)" is inconsistent with the results presented in Figures 5 D and E regarding the effect of the treatment with RSV and QCT alone at a concentration of 6.25 μM on a percentage of PI 390 positive cells.
Author Response
ANSWER TO REVIEWER
Prevention of 7-ketocholesterol-induced overproduction of reactive oxygen species, mitochondrial dysfunction and cell death with major nutrients (polyphenols, ω3 and ω9 unsaturated fatty acids) of the Mediterranean diet on N2a neuronal cells
Yammine A et al. – R2
The answers are underlined in yellow
The authors addressed most comments, and the manuscript has improved as a result. However, there is still a confusing description of results.
Question: The commentary that "Noteworthy, when used singularly, polyphenols (QCT, API, RSV) and fatty acids (ALA, 397 EPA, DHA, OA) as well as α-tocopherol have no effect on plasma membrane integrity (Figures 5-6)" is inconsistent with the results presented in Figures 5 D and E regarding the effect of the treatment with RSV and QCT alone at a concentration of 6.25 μM on a percentage of PI 390 positive cells.
Answer: this has been done; we agree with this comment. The change has been made and we apologize for the incomplete description regarding the effects of RSV and QCT at the plasma membrane level. The description now takes into account the small but significant increase in plasma membrane permeability observed with RSV and QCT at 6.25 µM. As a result, the discussion sentence addressing the toxicity of RSV and QCT has also been modified.